# Variations in grain yield and nutrient status of different maize cultivars by application of zinc sulfate

Juan Xin[1]*, Ning Ren[2], Xueling Hu[3], Jin Yang[3]

1 National Institute of Central Cities, Zhengzhou Normal University, Zhengzhou, China, 2 Agriculture Rural Development Service center of Xun Xian, Hebi, China, 3 College of Life Science, Zhengzhou Normal University, Zhengzhou, China

* xinjuan0707@163.com

**Data Availability Statement:** All relevant data are within the paper and its Supporting information files.

## Abstract

Although maize is sensitive to zinc (Zn) deficiencies, the responses of maize cultivars to the foliar application of Zn sulfate ($ZnSO_4$) may vary significantly. Here, we quantified the responses of grain yields and nitrogen (N), phosphorus (P), and potassium (K) absorption tto $ZnSO_4$ using 22 modern maize cultivars. The results revealed that 40.9% of the cultivars were not affected by foliar $ZnSO_4$, whereas only 45.5% of the cultivars responded positively to$ZnSO_4$, which was evidenced by increased grain numbers and shortened bald tip lengths. The impact of Zn fertilizer might be manifested in the dry biomass, from the 8-leaf stage (BBCH 18). For Zn-deficiency resistant cultivars, the foliar application of $ZnSO_4$ enhanced N accumulation by 44.1%, while it reduced P and K absorption by 13.6% and 23.7%, respectively. For Zn-deficiency sensitive maize cultivars, foliar applied $ZnSO_4$ improved the accumulation of N and K by 27.3% and 25.0%, respectively; however, it lowered their utilization efficiency. Hence, determining the optimized application of Zn fertilizer, while avoiding Zn toxicity, should not be based solely on the level of Zn deficiency in the soil, but also, take into consideration the sensitivity of some cultivars to Zn, Furthermore, the supplementation of Zn-deficiency sensitive maize cultivars with N and K is key to maximizing the benefits of Zn fertilization.

## Introduction

Deficiencies in microelements encompassing zinc (Zn), boron, iron, manganese, copper, and molybdenum are prevalent in poor soils on a global scale [1]. Worldwide, almost 49% of soils are Zn-deficient, whereas China and India lack the most soil resident trace elements across Asia, with China's soil Zn deficiency at ~51% [2, 3]. The main reasons for Zn-deficient soils in northern China are mainly in calcareous soils, because soil available Zn is negatively correlated with pH [4]. At pH levels of from 6 to 7 the chemical solubility of Zn is reduced to 1/30 of the original, as $Zn^{2+}$ is precipitated as $ZnCO_3$ and $Zn_5(CO_3)_2(OH)_2$ [5]. Moreover, intensive farming takes a huge amount of Zn and few farmers pay attention to supply it, resulting in soil Zn deficiency [2].

**Funding:** This work was financially supported by the special fund for doctoral research startup, Zhengzhou Normal University (No. 2018-702355).

**Competing interests:** The authors have declared that no competing interests exist.

**Abbreviations:** KHI, K harvest index; NHI, N harvest index; PHI, P harvest index; TKA, total K accumulation; TNA, total N accumulation; TPA, total P accumulation; Zn, deficiency non-sensitive type, Type N; Zn, deficiency resistant type, Type R; Zn, deficiency sensitive type, Type S.

Zn is an essential microelement for human health, and its deficiency leads to decrease in body weight, vision, cognition, and immunity [6]. Zn is also a necessary microelement for the growth and development of crop plants [7], as it promotes the synthesis of tryptophan from benzpyrole and serine, which is the synthetic precursor of indole acetic acid; thus, its deficiency slows plant growth and reduces dry matter weight. Moreover, it is also an essential component of many enzymes that affect the physiological metabolism and morphogenesis of plants, such as carbonic anhydrase, which is involved in foliar photosynthesis [8–10].

Dominic et al. [11] pointed out that maize grain yield and grain Zn density responses to Zn fertilizer application by up to 17 and 25% through meta-analysis from 67 publications. When plants show Zn deficiency, they require supplementation with Zn fertilizer, which may be applied either directly to the soil, via seed dressing, or foliarly [12]. Higher pH and $CaCO_3$ content of soils in northern China reduced the Zn availability on calcareous soils [13]. Because Zn has poor soil mobility and does not match the spatial distribution of roots, farmers typically select foliar spray to shorten the transport distance of Zn from soil to grain [14, 15]. It was recently documented that the foliar application of Zn sulfate ($ZnSO_4$) is a simple approach for quickly correcting the nutritional status of plants [16]. Further, the application foliar ($ZnSO_4$) spray is typically carried out during the early stage of crop growth, as the effectiveness of spraying at later stages is poor [17].

Since maize is an important grain, feed, and industrial crop in China, its stable production is of great significance for the country's food security. Maize is very sensitive to Zn deficiency, the lack of which decreases the surface areas of leaves and reduces chlorophyll content, with the formation of white leaves in cases of severe Zn deficiency [18]. Additionally, the photosynthetic capacities of maize leaves are reduced, and the development of yield components is also affected by Zn deficiencies, which are manifest as the reduced number of kernels per cob and grain weight [19, 20]. In Northern China, the application of ($ZnSO_4$) enhanced the average maize yield by 14.4%. Nevertheless, some farmers reflected that the resulting yield increases following the application of zinc fertilizer in maize were not always satisfactory, which was mainly dependent on whether the maize was sensitive to Zn deficiencies [21]. There are significant differences in the sensitivity of different genotypes of the same crop to Zn deficiency, which can be divided into zinc deficiency sensitive type and zinc deficiency insensitive types [22]. Usually, varieties with large root absorption area and strong absorption capacity are Zn efficient varieties, which are not closely related to the soil available zinc content [23]. The level of Zn efficiency in plants not only depends on the amount of Zn required, but mainly on the distribution efficiency of Zn in the parts of the plant with the highest demand [24]. Some genes responsible for Zn uptake and transport across membranes (*ZmZIP3*, *ZmHMA3*, *ZmHMA4*) were identified, which form a sophisticated network to regulate the uptake, translocation, and redistribution of Zn [25]. Besides genotypes, the application of Zn may impact the absorption and utilization of other elements, and the interactions of Zn with N, P, K have been reported in many crops [26, 27].

The objectives of this study were to: (1) screen the maize varieties that are sensitive and insensitive to Zn deficiency; (2) elucidate the physiological mechanism of zinc affecting corn yield from the perspectives of leaf senescence and macronutrient (NPK) absorption. The findings of this study provide a theoretical basis for rational spraying of Zn fertilizer on maize.

## Materials and methods

### Site description

Field experiments were conducted in Changge county (34°27′N 113°34′E; 102 m.a.s.l.), of Henan Province, in Central China. This field experiment was approved with Xuchang

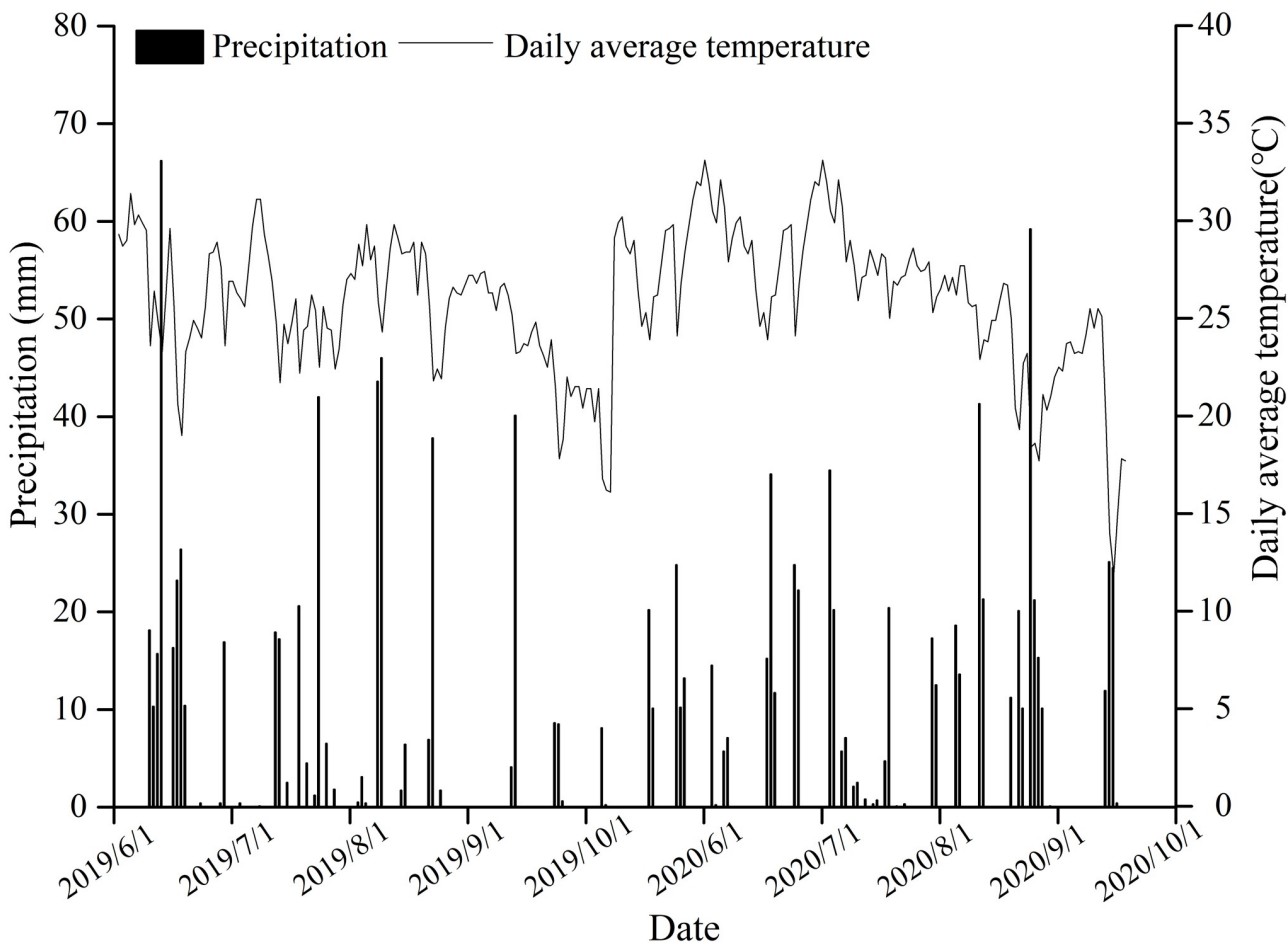

**Fig 1. Temperature and precipitation measurements during the maize-growing season.**

Agricultural Technology Extension Station. During the maize growing season (June–October, 2020–2021), the average daily air temperature and total precipitation were 30.4˚C and 537.3 mm (2020) and 31.2˚C and 647.2 mm (2021), respectively (Fig 1). Due to sufficient precipitation, we did not carry out additional irrigation during the maize growing season. Prior to the experiments, soil samples extracted from the upper 20-cm layer were collected for chemical analyses. The soil type was fluvoaquic soil (pH 7.8), with an organic matter content of 20.03 g $kg^{-1}$ (0–20 cm), total N of 1.45 g $kg^{-1}$, $NO_3^{-}$-N 12.49 mg $kg^{-1}$, $NH_4^{+}$-N 3.17 mg/kg, available P (Olsen-P) of 18.67 mg $kg^{-1}$, available K ($NH_4OAc$-K) of 157.32 mg $kg^{-1}$, and available Zn (DTPA-Zn) of 0.40 mg $kg^{-1}$ (Zn deficient soil) [14].

## Experimental design and management

The zinc sulfate ($ZnSO_4$) treatments were arranged in a split-plot design, with three replications. The primary plots used for the experiments consisted of 22 maize cultivars, which were extensively cultivated in the Huang-Huai-Hai Plain, with the maize cultivars listed in S1 Table. The collection of plant material in this article complied with relevant institutional, national, and international guidelines and legislation. Each plot was divided into two subplots: one was sprayed with 1350 g $ZnSO_4\cdot7H_2O$ at 450 L solution per hectare (+Zn), which is the

recommended concentration in China [14], while the other plot was treated with distilled water (CK) (S1 Fig). Tween-20 [(polyethylene glycol sorbitan monolaurate); Aladdin Industrial Corporation, Shanghai, China] was also added to the solutions to serve as a surfactant. The $ZnSO_4$ solution was applied via a knapsack electric sprayer at the 8-leaf stage (BBCH 18), at between 17:00 and 18:00, which was repeated approximately 10 days later.

The seeds were mechanically sown on June 3, at a hill spacing of $0.60 \times 0.27$ m, with 61,725 plants ha$^{-1}$, with the dimensions of each plot being $4 \times 10$ m. The seed (about 3–5 cm depth) and fertilizer (about 8–10 cm depth) were applied at the same time through spoon wheel type maize simple grain sowers (2BYFSF-4, Hebei Nonghaha Machinery Group Co., Ltd, Shijiazhuang, China). Nitrogen [180 kg(N) ha$^{-1}$] in the form of urea was applied in two equal splits with 50% at basal, and 50% at the stem elongation stage (BBCH 31). Phosphorus [90 kg($P_2O_5$) ha$^{-1}$ as triple superphosphate] and potassium [90 kg($K_2O$) ha$^{-1}$ as potassium chloride] was applied as a basal dose. The fertilization formula and dosage of this study referred to Tian et al. [28]. Field sites took the wheat–maize rotation system during the experimental period. The fertilizer rate of wheat growing season was the same with maize growing season at each plot. Nicosulfuron and atrazine were applied at the 3-leaf stage (BBCH 13) to control weeds, whereas thiophanate-methyl and lambda-cyhalothrin were applied at the 8-leaf stage (BBCH 18) to prevent diseases and insects.

## Maize type classification

The effects on the yields of different maize cultivars following the application foliar $ZnSO_4$ varied significantly. The 22 maize cultivars were divided into three types based on the rate of increase (Table 1; Fig 2): Zn-deficiency sensitive type (Type S), Zn-deficiency non-sensitive type (Type N), and Zn-deficiency resistant type (Type R).

The between-groups linkage method in hierarchical cluster analysis (SPSS19.0 software, Chicago, IL, USA) was used for maize variety classification, with an interval of Squared Euclidean distance. In 2020, the yield increase rate range of S-type maize was 9.6% to15.3%, the range of N-type maize was 1.5% to 7.2%, and the range of R-type maize was -7.9% to -5.4%. In 2021, the yield increase rate range of S-type maize was 10.4% to13.3%, the range of N-type maize was 1.1% to 7.1%, and the range of R-type maize was -8.8% to -6.4%. During both years, the N-type maize accounted for the largest proportion, reaching 54.5%.

**Table 1. Classification of 22 maize cultivars based on increased yield rate through the application of foliar $ZnSO_4$.**

| Type | Yield increase rate (%) | | Count | Percentage (%) |
|---|---|---|---|---|
| | Range | Mean | | |
| 2020 | | | | |
| S | 9.6 ~ 15.3 | 11.6 | 6 | 27.3 |
| N | 1.5 ~ 7.2 | 3.6 | 12 | 54.5 |
| R | -7.9 ~ -5.4 | -6.6 | 4 | 18.2 |
| 2021 | | | | |
| S | 10.4 ~ 13.3 | 11.7 | 6 | 27.3 |
| N | 1.1 ~ 7.1 | 4.0 | 12 | 54.5 |
| R | -8.8 ~ -6.4 | -7.2 | 4 | 18.2 |

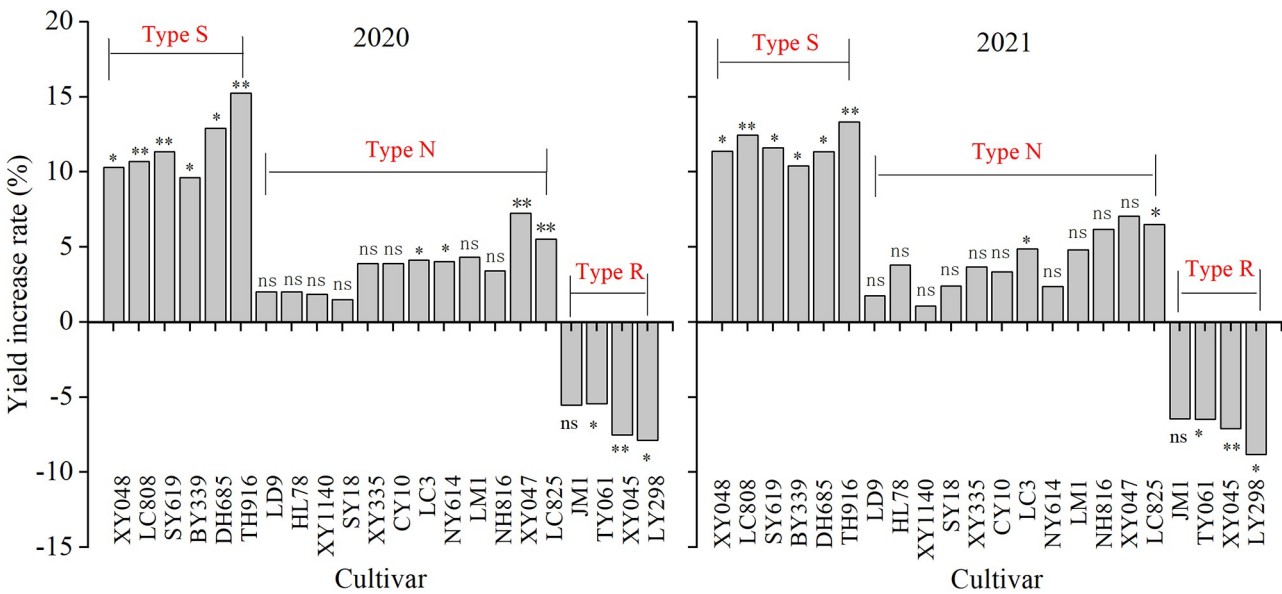

**Fig 2. Effects of foliar ZnSO$_4$ application on yield rates for different maize cultivars.** Significant yield differences after the addition of Zn fertilizer in the same maize cultivar are indicated by *($P < 0.05$), **($P < 0.01$) and ns (no significance).

## Chlorophyll, photosynthesis and MDA measurements

Chlorophyll meter readings were obtained using a hand-held dual-wavelength meter (SPAD-502, Minolta Camera Co., Ltd., Japan) from the mid-point of the ear leaf at the 8-leaf (BBCH 18) and tasseling stage (BBCH 65). The ear leaves of ten consecutive plants in one of the central rows were selected, and their SPAD values were measured in the morning (08:00–11:00 h).

At the tasseling (BBCH 65) stage, eight tagged ear leaves from each plot were selected in the morning (10:00–11:30 h). The ear leaves were used to quantify the net photosynthetic rate ($P_n$) using a portable photosynthesis apparatus, equipped with a red and blue LED light source leaf chamber (Li-6400, Li-Cor, Lincoln, NE, USA). For each measurement, the middle section of a leaf was enclosed in the leaf chamber and the $P_n$ was recorded following equilibration to a steady state.

At the grain filling stage (BBCH 75), malondialdehyde (MDA) concentration was determined according to the methods described by [29]; 0.5 g of fresh leaves was homogenized, containing 10% trichloroacetic acid and 0.5% 2-thiobarbituric acid, and then heated in boiling water for 20 min. After rapid cooling with ice, the mixture was centrifuged at 3,000 rpm for 10 min. The absorbance of the supernatant was measured at 450, 532 and 600 nm using a spectrometer (UV-1800, Shimadzu, Japan). The concentration of MDA was calculated as: MDA [$\mu$mol L$^{-1}$] = $6.45 \times (OD_{532} - OD_{600}) - 0.56 \times OD_{450}$.

## Plant sampling and nutrient measurements

At the 8-leaf (BBCH 18), tasseling (BBCH 65) and physiological maturity (BBCH 87) stages, five consecutive maize plants from each plot were sampled, and then dissected into leaf, stem, cob and grain. The fresh samples were oven dried at 105°C for 30 min., and then at 75°C, until a constant weight was achieved. The yield of corn grains was adjusted to 13% moisture content. The plant materials were ground to pass through a 1-mm mesh screen, and then digested by H$_2$SO$_4$ and H$_2$O$_2$. The N and P concentrations of the digested samples were determined

using an automated continuous flow analyzer (Seal, Norderstedt, Germany). The K concentrations of the digested samples were determined with a flame photometer FP-640 (Precision Instrument Co., Ltd., Shanghai, China).

## Data analysis

The formula for calculating the absorption and utilization efficiency parameters of the three nutrients (N, P, and K) is similar. The example of N is given as follows:

$$\text{TNA}\left[\text{kg ha}^{-1}\right] = \text{plant N concentration}\left[\text{kg kg}^{-1}\right] \times \text{plant dry matter}\left[\text{kg ha}^{-1}\right];$$

$$\text{NA in } 100\,\text{kg of grains}[\text{kg}] = \text{TNA}\left[\text{kg ha}^{-1}\right]/\text{corn grain yield}\left[\text{kg ha}^{-1}\right] \times 100\,[\text{kg}];$$

$$\text{NHI}[\%] = \text{grain N accumulation}\left[\text{kg ha}^{-1}\right]/\text{TNA}\left[\text{kg ha}^{-1}\right] \times 100[\%];$$

where TNA is the total N accumulation, NHI is the N harvest index.

The statistical analyses focused on the effects of relating factors in the split-plot design, as well as their interactions with various parameters of agronomic crop traits and nutrient utilization efficiencies. Therefore, all data were subjected to an *ANOVA* using the general linear model procedure in SPSS 19.0 software (Chicago, IL, USA), and the mean values of the treatments were compared on the basis of the least significant difference test (LSD). The graphs were plotted using the Origin 9.0 program.

## Results

### Yields and yield components of different maize cultivars

Data associated with the grain yields and their components are shown in Table 2. The results revealed that the grain yields and grain numbers per panicle of different maize cultivars were in the order of S-type < N-type < R-type under no-Zn treatment, while bald tip length showed the opposite trend. The application of foliar $ZnSO_4$ significantly increased the grain yields of S-type maize, whilst significantly decreased the grain yields of the R-type maize. The increased yield of the S-type maize was primarily attributed to the higher grain numbers, whereas the decreased yield of the R-type maize was related to both the reduced grain number and 1000-grain weight. The application of $ZnSO_4$ decreased the bald tip length, which emerged as a significant effect only for the S-type maize. The grain weight, as well as maize types were shown to be unaffected by the application of foliar $ZnSO_4$.

### Leaf photosynthetic characteristics of different maize cultivars

At the BBCH 18 stage, the leaf SPAD value of type-S, type-N, and type-R maize was significantly increased by 9.2%, 7.0%, and 7.1%, respectively, following the addition of $ZnSO_4$ during 2020 growing season; whilst, the leaf SPAD value of type-S, type-N, and type-R maize was significantly increased by 8.1%, 8.5%, and 7.8% during 2021 growing season, respectively (Fig 3). At the BBCH 65 stage, only the SPAD value of the S-type maize was significantly enhanced following the application of $ZnSO_4$. The foliar application of $ZnSO_4$ increased of the $P_n$ of ear leaves significantly in the S-type (14.3%) and N-type maize (8.1%), while no obvious difference was found between CK and +Zn treatments in R-type maize (3.5%) (Fig 4).

**Table 2. Effects of foliar $ZnSO_4$ application on yields and their components in different maize cultivars.**

| Type | Treatments | Grain yield (t ha$^{-1}$) | Grains number per cob | 1000 grains weight (g) | Bald tip length (cm) |
|---|---|---|---|---|---|
| 2020 | | | | | |
| S | CK | 10.84 ± 0.37 | 509.8 ± 5.8 | 343.1 ± 2.8 | 1.87 ± 0.15 |
| | +Zn | 12.10 ± 0.26** | 546.3 ± 9.1** | 355.1 ± 6.5$^{ns}$ | 1.35 ± 0.21** |
| N | CK | 11.64 ± 0.24 | 544.2 ± 6.4 | 350.0 ± 2.0 | 1.28 ± 0.16 |
| | +Zn | 12.07 ± 0.44$^{ns}$ | 550.8 ± 7.9$^{ns}$ | 354.3 ± 6.7$^{ns}$ | 0.98 ± 0.08* |
| R | CK | 13.26 ± 0.20 | 576.7 ± 12.4 | 362.2 ± 8.4 | 0.81 ± 0.12 |
| | +Zn | 12.39 ± 0.31* | 563.8 ± 10.2$^{ns}$ | 352.8 ±2.5$^{ns}$ | 0.69 ± 0.08$^{ns}$ |
| Zn | | ** | * | ns | * |
| Type | | ** | ** | ns | ** |
| Zn×Type | | ** | ** | ns | * |
| 2021 | | | | | |
| S | CK | 9.64 ± 0.12 | 400.9 ± 9.4 | 305.8 ± 3.3 | 1.55 ± 0.22 |
| | +Zn | 11.77 ± 0.23** | 440.6 ± 10.2** | 312.9 ± 4.8$^{ns}$ | 1.28 ± 0.14** |
| N | CK | 10.4 ± 0.27 | 434.4 ± 14.4 | 303.3 ± 1.8 | 1.34 ± 0.21 |
| | +Zn | 10.9 ± 0.26$^{ns}$ | 446.6 ± 11.2$^{ns}$ | 315.7 ± 6.1$^{ns}$ | 0.89 ± 0.13** |
| R | CK | 11.74 ± 0.21 | 465.4 ± 16.2 | 303.8 ± 1.4 | 0.62 ± 0.18 |
| | +Zn | 10.88 ± 0.25* | 456.2 ± 10.7$^{ns}$ | 311.0 ± 2.8$^{ns}$ | 0.41 ± 0.11$^{ns}$ |
| Zn | | ** | * | ns | ** |
| Type | | ** | ** | ns | ** |
| Zn×Type | | ** | * | ns | * |

Values are means ± SD ($n$ = 3). Significant differences between these two treatments for the same maize type are indicated by

*($P < 0.05$),

**($P < 0.01$) and

$^{ns}$ (no significance).

## Leaf lipid peroxidation of different maize cultivars

The foliar application of $ZnSO_4$ decreased the MDA content of ear leaves significantly in the S-type, while increased significantly in R-type maize (Fig 5). And no obvious MDA concentration difference was found between CK and +Zn treatments in N-type maize.

## Dry matter accumulation of different maize cultivars

At the BBCH 18 and BBCH 65 stages, the dry matter of the R-type and N-type maize was significantly higher than that of the S-type under a no-Zn condition (Fig 6). For the S-type or R-type maize, the Zn-induced variations in the dry matter were gradually increased with the development of the growth process. At the BBCH 87 stage, the dry matter of the S-type cultivars was significantly increased by 8.4% (2020) and 11.0% (2021), while, dry matter of the R-type maize was remarkably reduced by 10.9% (2020) and 12.2% (2021). For the N-type cultivars, the increasing effect was found only at the BBCH 18 stage during the 2020 growing season; however, no significant differences were observed in the subsequent reproductive period.

## Nutrient uptake and utilization of different maize cultivars

The TPA and TKA were statistically similar among the three types of maize; however, the TNA revealed the sequence of S-type > N-type > R-type (Table 3). The application of $ZnSO_4$ significantly increased the TNA of all maize types, particularly the R-type maize, where the

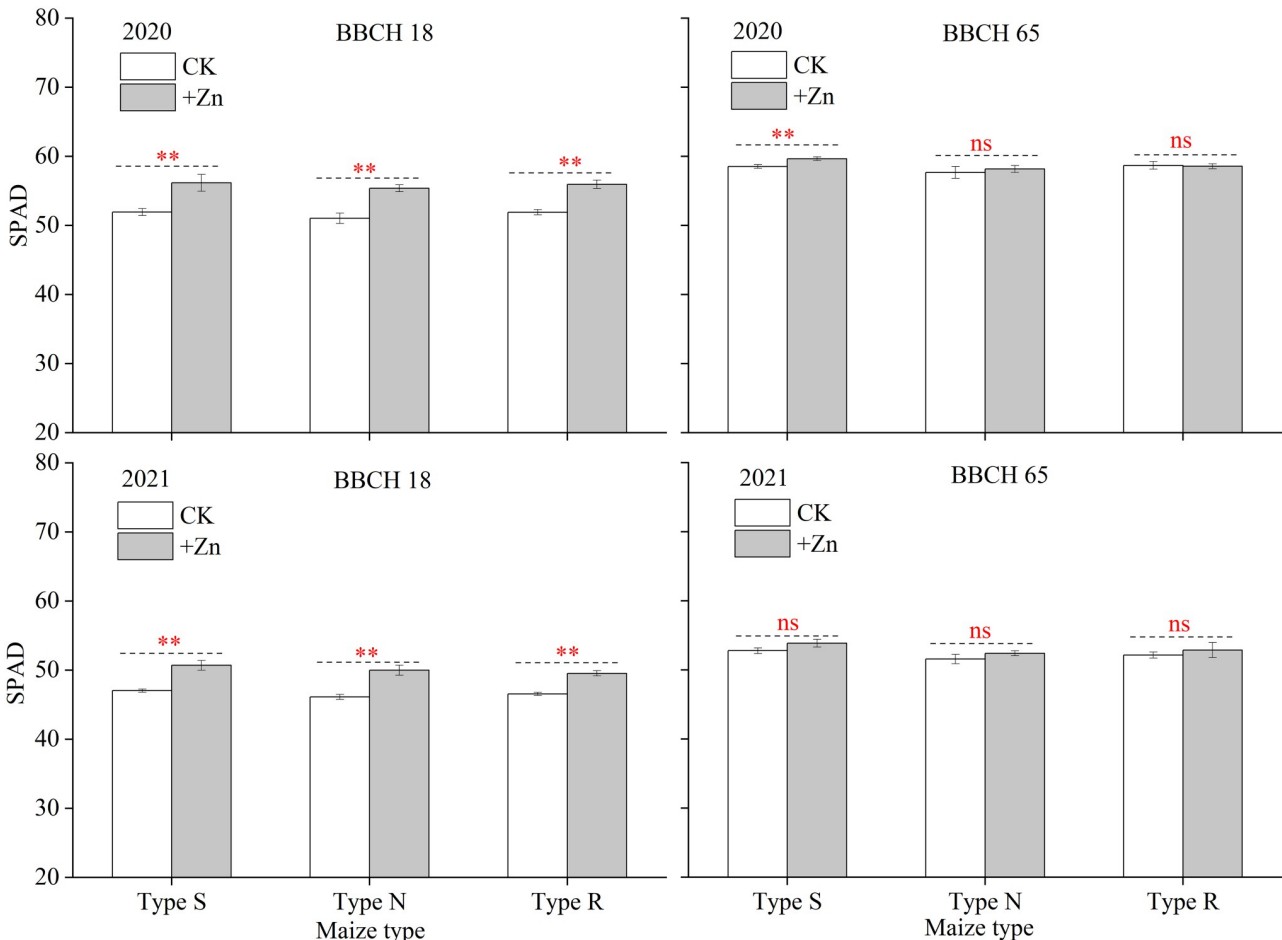

**Fig 3. Effects of foliar ZnSO₄ application on leaf SPAD values for different maize cultivars.** Significant differences between two Zn spraying treatments in the same maize type are indicated by *($P < 0.05$), **($P < 0.01$) and ns (no significance). Each bar represents the mean ± SD (n = 3).

TNA was increased by 44.2%. Accompanied with the increase of TNA in the R-type maize, the TPA and TKA decreased dramatically. In contrast to the R-type maize, the TKA in the H and N-type maize was enhanced following the addition of foliar ZnSO₄, while the TPA in the H and N-type maize was unaffected by foliar ZnSO₄. The NHI and KHI of all the maize types were significantly reduced following the application of Zn fertilizer, except for the KHI in the R-type maize. Maize treated with Zn increased the PHI, where the rate of increase in the S-type maize was highest (34.2%), followed by the N-type (31.7%), with the R-type having the lowest (22.4%).

## Relationship between photosynthetic characteristics, nutrient uptake and yield

The correlation figure showed that yield was positively correlated with leaf SPAD and dry matter (Fig 7). Although there was a significant positive correlation between dry matter and N, P, K uptake, nutrient uptake did not directly determine grain yield. In addition, TNA showed significant positive correlation with leaf SPAD, Pn and dry matter. The MDA had a significant negative correlation with most of the parameters.

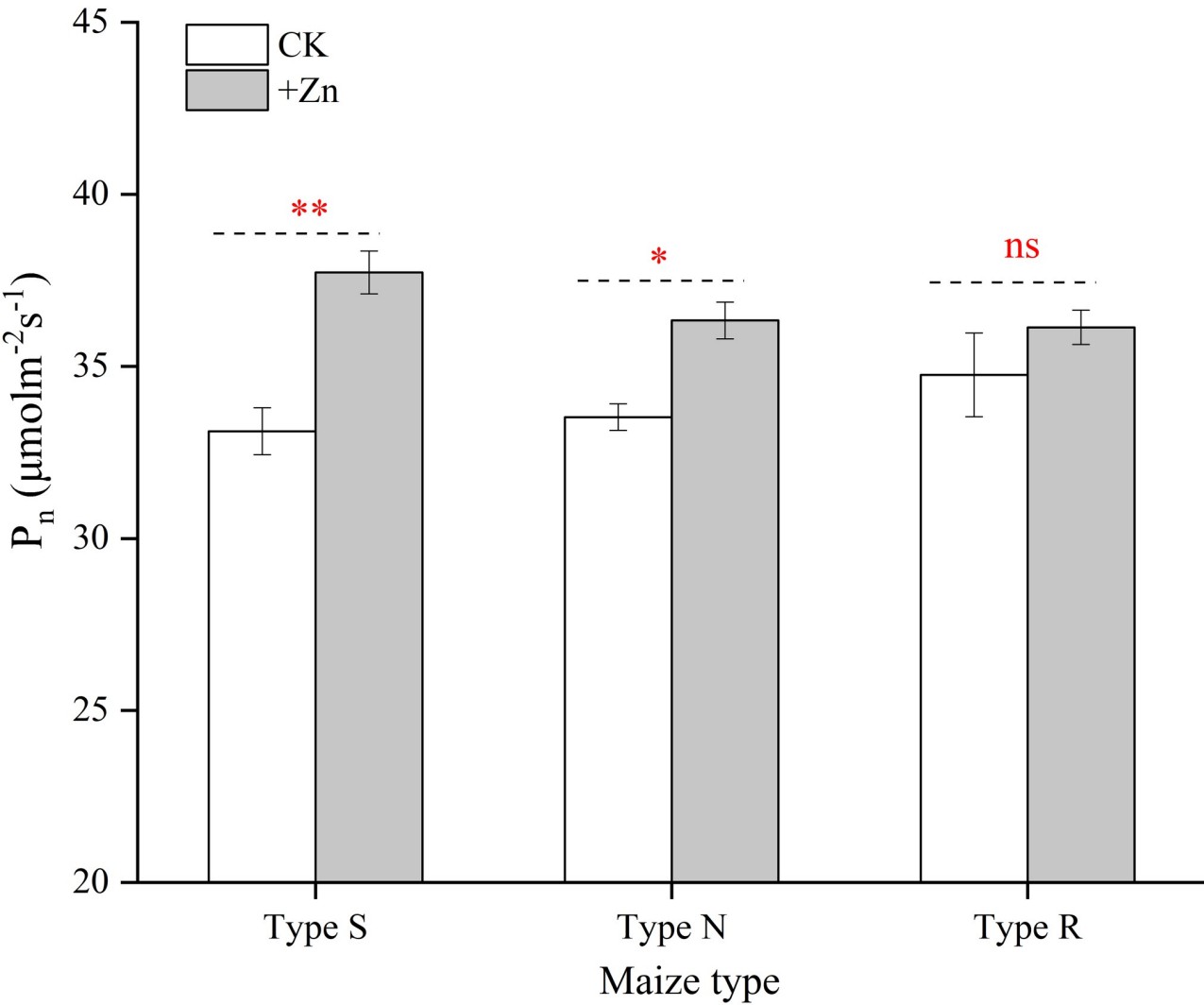

**Fig 4. Effects of foliar ZnSO$_4$ application on ear leaf photosynthetic parameters for different maize cultivars at grain filling stage.** Significant differences between two Zn spraying treatments in the same maize type are indicated by *($P < 0.05$), **($P < 0.01$) and ns (no significance). Each bar represents the mean ± SD (n = 3).

## Discussion

### Mechanism of Zn-induced increase in maize yield

Under Zn-deficient soils, the application of Zn fertilizer can increase the productivity and quality of cereal crops, vegetables, and fruits [12, 30, 31]. This was related to Zn induced increases in photosynthesis, auxin synthesis, and the activities of various enzymes [32, 33]. Zn activation capacity of roots, availability and diffusion capacity of Zn in soil could affect Zn uptake by plants [23]. Soil with high total zinc content may not necessarily have strong Zn supply capacity, as higher soil pH and CaCO$_3$ content in northern China reduce the effectiveness of soil Zn [13]. Generally, foliar Zn fertilizer has a better effect on improving crop grain yield and Zn content than soil Zn fertilizer application [34]. In the present study, foliar ZnSO$_4$ spray significantly increased the grain yields of 10 maize cultivars, which accounted for 45.5% of the

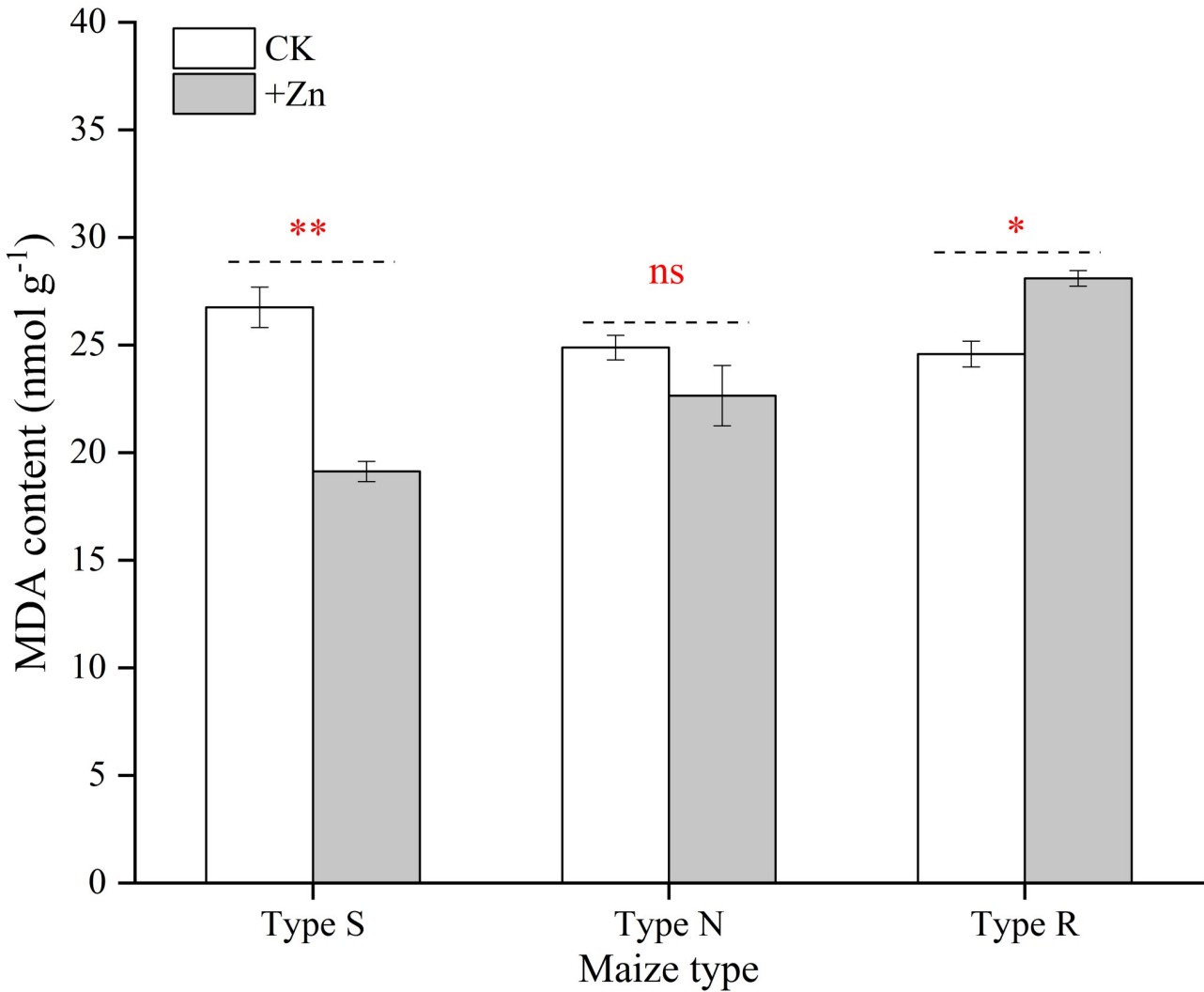

**Fig 5. Effects of foliar ZnSO$_4$ application on ear leaf MDA content for different maize cultivars at grain filling stage in 2021.** Significant differences between two Zn spraying treatments in the same maize type are indicated by *($P < 0.05$), **($P < 0.01$) and ns (no significance). Each bar represents the mean ± SD (n = 3).

total cultivars (Fig 2; Table 1). The increase in maize yield was primarily accompanied by higher grain numbers and additional grain weight (Table 1). Previous studies also found similar results for maize [35]. Further, our research revealed that ZnSO$_4$ spray shortened the maize bald tip length. This may be related to Zn spraying that enhanced the content of auxin and gibberellin in corn, which had a significantly negative relationship with the length of bald tip [36].

The dry matter of the 18 maize cultivars was enhanced following the addition of Zn fertilizer, while it increased their N and K uptakes (Table 3). The maize absorbed additional N and K; however, most of the N and K was retained in the straw to a higher degree, which resulted in producing the same amount of grain that required extra N and K (Table 3). Hence, with the application of foliar ZnSO$_4$, it was necessary to supply N and K fertilizer to achieve better production. When plants suffered from Zn-deficiency, the photosynthetic characteristics, photosynthetic pigment content and NR activity was decreased which was caused by the decline in

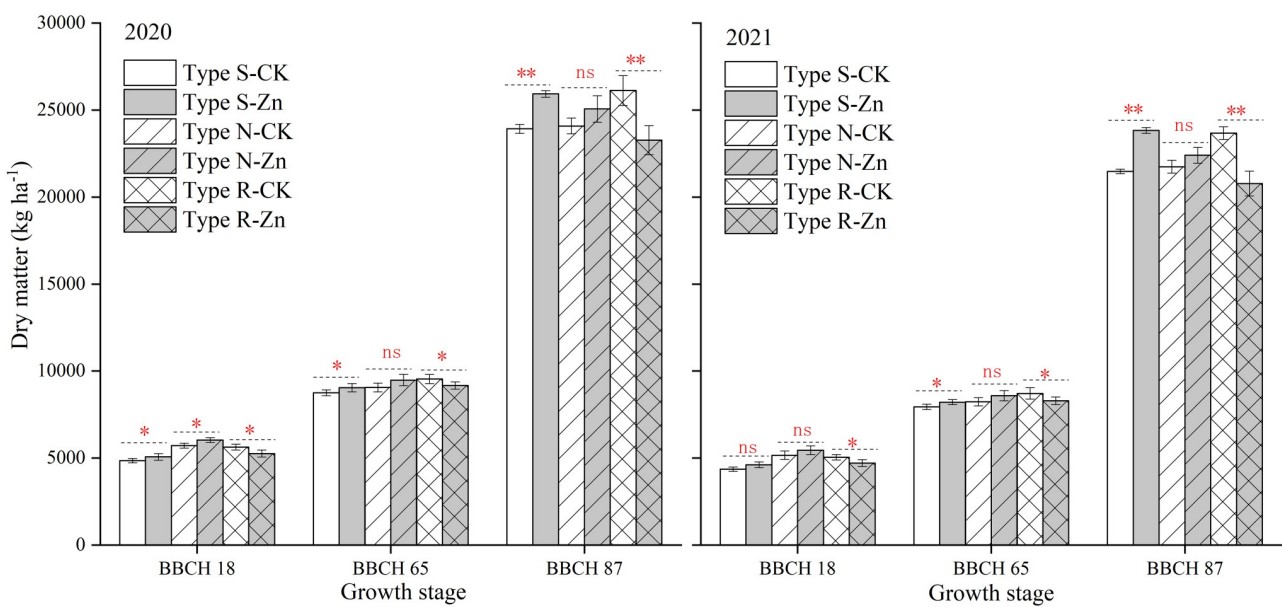

**Fig 6. Effects of foliar ZnSO4 application on aboveground biomass for different maize cultivars at different growth stages.** Significant differences between two Zn spraying treatments in the same maize type are indicated by *($P < 0.05$), **($P < 0.01$) and ns (no significance). Each bar represents the mean ± SD (n = 3).

carbonic anhydrase activity [37, 38]. In this study, the leaf SPAD and $P_n$ was increased after the addition of Zn, which further increased plant dry matter (Figs 3 and 4). Previous studies suggested that Zn deficiency causes the accumulation of reactive oxygen species and the oxidative degradation of the indole-3-acetic acid, but foliar Zn spraying reversed this phenomenon [39]. In this research, the application of foliar $ZnSO_4$ decreased the MDA content of ear leaf in type-S maize, protecting the chloroplast cell membrane structure (Fig 5). Both higher leaf photosynthetic characteristic and extended leaf function period enhanced the accumulation of plant biomass in this study. Antagonism between Zn and phosphorus, which have been reported for various crops in numerous research studies [40–42]. In the present study, the application of foliar $ZnSO_4$ suppressed P absorption, which verified the results of previous studies. However, it was interesting that more P was transferred from the vegetative components to reproductive organs under $ZnSO_4$ spraying (Table 3). This might have been due to decreased P absorption by roots, after which P was autonomously distributed more to the growth center [40]. There is no literature reporting that Zn can promote the transport of P.

## Differences in sensitivity of maize cultivars to Zn fertilizer

Cereal crops differ in their sensitivity to Zn deficiency. Rice, sorghum and maize belonged to Zn-sensitive, while barley, wheat, rye and oats were considered to be less sensitive to Zn [43]. Zn deficiency in soil-crop systems has become more prevalent in recent decades because of intensive farming, use of high yielding cultivars, and lack of Zn fertilization [44]. However, reports on increase in maize yields through the application of Zn are frequently documented, and the quantity of Zn fertilizer employed for maize production ranks first among all crops in China [45].

Interestingly, the responses of different genotypes of the same species to Zn have been observed to be inconsistent [21, 46]. For Zn-deficiency insensitve variety, the overexpression

**Table 3. Effects of foliar $ZnSO_4$ application on N, P, and K uptake and utilization for different types of maize cultivars.**

| Type | Treatments | TNA (kg ha$^{-1}$) | NHI (%) | TPA (kg ha$^{-1}$) | PHI (%) | TKA (kg ha$^{-1}$) | KHI (%) |
|---|---|---|---|---|---|---|---|
| 2020 | | | | | | | |
| S | CK | 264.1 ± 17.3 | 61.2±4.7 | 129.9±5.6 | 42.1±5.4 | 281.8 ± 20.2 | 13.8±1.2 |
| | +Zn | 336.2 ± 24.2** | 49.9±2.1* | 120.8±11.8$^{ns}$ | 56.5±6.1* | 352.2 ± 10.7** | 10.9±1.9* |
| N | CK | 258.5 ± 19.0 | 61.6±2.4 | 129.9±5.5 | 36.9±2.1 | 296.7 ± 10.8 | 14.3±0.5 |
| | +Zn | 308.5 ± 14.0** | 54.1±1.4** | 122.0±14.5$^{ns}$ | 48.6±5.6* | 331.5 ± 10.3* | 10.3±0.2** |
| R | CK | 222.9 ± 17.5 | 63.1±2.3 | 127.4±7.8 | 39.7±3.3 | 345.6 ± 18.7 | 15.3±1.3 |
| | +Zn | 321.3 ± 9.0** | 51.8±1.8** | 110.0±8.0* | 48.6±5.0* | 263.8 ± 19.1** | 14.2±0.7$^{ns}$ |
| Zn | | ** | ** | ns | ** | ns | ** |
| Type | | ** | ns | ns | ns | ns | ** |
| Zn×Type | | ** | ns | ns | ns | ns | * |
| 2021 | | | | | | | |
| S | CK | 218.7 ± 17.0 | 60.9 ± 6.9 | 110.4 ± 18.6 | 41.0 ± 4.9 | 236.7 ± 11.2 | 12.9 ± 0.7 |
| | +Zn | 303.9 ± 7.7** | 44.9 ± 1.5** | 109.9 ± 31.1$^{ns}$ | 49.3 ± 3.5* | 315.2 ± 5.7** | 9.8 ± 0.2* |
| N | CK | 217.2 ± 13.8 | 61.2 ± 3.2 | 110.2 ± 3.6 | 39.6 ± 4.2 | 252.6 ± 11.3 | 13.0 ± 0.8 |
| | +Zn | 259.6 ± 5.0** | 54.1 ± 1.7* | 105.2 ± 12.9$^{ns}$ | 50.4 ± 6.0* | 283.2 ± 9.1* | 9.7 ± 0.4** |
| R | CK | 190.6 ± 3.2 | 77.0 ± 0.9 | 108.7 ± 4.3 | 36.4 ± 3.7 | 286.5 ± 23.1 | 12.8 ± 1.0 |
| | +Zn | 269.1 ± 12.6** | 51.0 ± 1.4** | 93.5 ± 5.8* | 45.2 ± 3.9* | 233.8 ± 19.9* | 13.7 ± 1.7$^{ns}$ |
| Zn | | ** | ** | ns | * | ns | * |
| Type | | ** | ns | ns | ns | ns | ** |
| Zn×Type | | ** | ns | ns | ns | ns | * |

TNA, total N accumulation; NHI, N harvest index; TPA, total P accumulation; PHI, P harvest index; TKA, total K accumulation; KHI, K harvest index. Values are means ± SD ($n$ = 3). Significant differences between these two treatments for the same maize type are indicated by

*($P < 0.05$),

**($P < 0.01$) and

$^{ns}$ (no significance).

of *ZmZIP3* and Zm*HMA4* was found under the Zn deficiency conditions, which facilitated the transport of Zn in the root-to-shoot and improved Zn utilization efficiency [25]. In the present study, about a quarter of the maize cultivars exhibited a decline in grain yields with the addition of foliar $ZnSO_4$ in this study. The grain yields of four maize cultivars (R-type) were higher than others (S-type and N-type) without $ZnSO_4$ spraying. Since the R-type maize was highly resistant to soil Zn deficiencies, the extra Zn applied through spraying might cause them to suppress Zn. Compared with other heavy metals, Zn is less toxic; however, it might cause certain damage to plants if excessive [47, 48]. In the present study, foliar $ZnSO_4$ spray reduced the dry matter of R-type maize at the 8-leaf stage, and the rate of decline was exacerbated with growth (Fig 6). Although the chlorophyll content was temporarily increased at the 8-leaf stage, this did not translate to an upsurge in photosynthesis. In fact, high Zn stress might have acted to hinder the electron transport chain, which in turn reduced the photosynthetic rate of leaves [49, 50]. In our study, the application of Zn did not affect the net photosynthetic rate of type-R maize, but significantly increased their MDA content at the grain filling stage (Fig 5). Senescence caused by zinc stress during late growth stage may be an important reason for yield decline in type-R maize (Fig 7). In contrast to S-type and N-type maize, the K concentration of R-type maize was evidently decreased following the addition of foliar $ZnSO_4$. Additionally, the N concentration of R-type maize was higher than that of H/N-type maize with the application of $ZnSO_4$. The N and K imbalance of R-type maize, particularly the excessive N/K ratio, likely initiated a reduction in production [51].

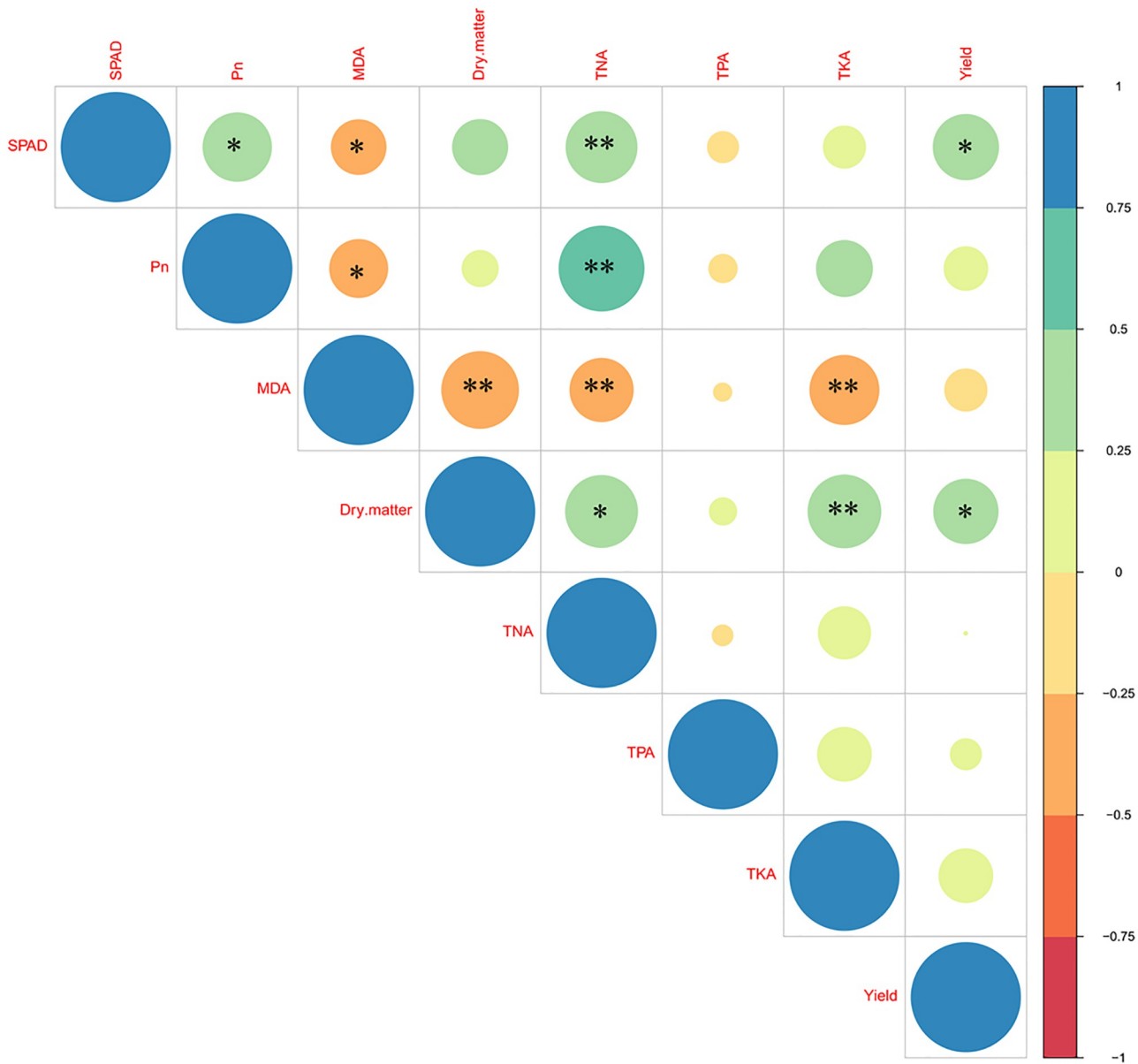

**Fig 7. Correlation between photosynthetic characteristics, nutrient uptake and yield.** TNA, total N accumulation; NHI, N harvest index; TPA, total P accumulation; PHI, P harvest index; TKA, total K accumulation; KHI, K harvest index. * and ** indicate significant differences at the levels of 0.05 and 0.01, respectively.

## Conclusions

In this study, 6 Zn-deficiency sensitive varieties, 12 Zn-deficiency non-sensitive varieties, and 4 Zn deficiency resistant varieties types were screened from 22 maize varieties. The responses of maize cultivars to the foliar application of $ZnSO_4$ may vary greatly, thus the application of Zn fertilizer cannot be based only on the soil Zn deficiency level, but should also consider the sensitivity of the cultivar to Zn. For Zn-deficiency sensitive varieties, the application of Zn increased the absorption of N and K nutrients, enhancing photosynthesis, delaying leaf senescence, further increasing grain number and weight. For Zn-deficiency resistant varieties, their

Zn demand was very small, and foliar application of Zn reduced the absorption of P and K nutrients in maize, accelerating leaf senescence, which in turn reduces yields.

## Supporting information

**S1 Fig. Field layout of the split-plot design experiment.**
(DOCX)

**S1 Table. The name for selected 22 modern maize cultivars.**
(DOCX)

**S1 Data. Raw data.**
(XLSX)

## Acknowledgments

We are grateful to Dr. Yang Wang for his guidance on our maize cultivation and field management.

## Author Contributions

**Conceptualization:** Juan Xin.

**Investigation:** Ning Ren, Xueling Hu, Jin Yang.

**Methodology:** Jin Yang.

**Writing – original draft:** Juan Xin, Ning Ren.

**Writing – review & editing:** Juan Xin, Xueling Hu, Jin Yang.

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
