## [Decision Letter · Decision Letter 0]

11 Jul 2023

PONE-D-23-07689Variations in grain yield and nutrient status of different maize cultivars by exogenous application of zinc sulphatePLOS ONE

Dear Dr. Xin,

Thank you for submitting your manuscript to PLOS ONE. After careful consideration, we feel that it has merit but does not fully meet PLOS ONE’s publication criteria as it currently stands. Therefore, we invite you to submit a revised version of the manuscript that addresses the points raised during the review process.

The manuscript requires additional work before being considered for possible acceptance. In particular, the reviewers emphasize the need to improve the Materials and Methods and Discussion sections. They also point out specific issues that must be resolved in discussing the results.

We look forward to receiving your revised manuscript.

Kind regards,

Adalberto Benavides-Mendoza, Ph.D.

Academic Editor

PLOS ONE

Journal Requirements:

“The authors are thankful to the project, “Effects of stress on wheat seed germination and seedling growth metabolism with different nitrogen content” funded by the special fund for doctoral research startup, Zhengzhou Normal University for the study”

7. PLOS requires an ORCID iD for the corresponding author in Editorial Manager on papers submitted after December 6th, 2016. Please ensure that you have an ORCID iD and that it is validated in Editorial Manager. To do this, go to ‘Update my Information’ (in the upper left-hand corner of the main menu), and click on the Fetch/Validate link next to the ORCID field. This will take you to the ORCID site and allow you to create a new iD or authenticate a pre-existing iD in Editorial Manager. Please see the following video for instructions on linking an ORCID iD to your Editorial Manager account: https://www.youtube.com/watch?v=_xcclfuvtxQ

8.  Please upload a copy of Supplementary Table S1 which you refer to in your text on page 5.

Additional Editor Comments:

The manuscript requires additional work before being considered for possible acceptance. In particular, the reviewers emphasize the need to improve the Materials and Methods and Discussion sections. They also point out specific issues that must be resolved in discussing the results.

Reviewers' comments:

Reviewer's Responses to Questions

**Comments to the Author**

1. Is the manuscript technically sound, and do the data support the conclusions?

Reviewer #1: No

Reviewer #2: Yes

Reviewer #3: Partly

2. Has the statistical analysis been performed appropriately and rigorously? 

Reviewer #1: Yes

Reviewer #2: Yes

Reviewer #3: Yes

3. Have the authors made all data underlying the findings in their manuscript fully available?

Reviewer #1: Yes

Reviewer #2: Yes

Reviewer #3: Yes

4. Is the manuscript presented in an intelligible fashion and written in standard English?

Reviewer #1: Yes

Reviewer #2: Yes

Reviewer #3: Yes

5. Review Comments to the Author

Reviewer #1: The manuscript shows interesting results that may be useful for the scientific community and for corn farmers. However, I believe that some aspects of it should be improved. Here are some recommendations:

L28. The keywords must be different from those of the title.

L38. Does what is mentioned occur in all types of soil?

L44. Mainly in alkaline pH soils.

L87. Why is it thought that the depth of soil sample? Consider that the roots of corn explore much more than 20 cm.

L103. Are foliar applications done between 5:00 p.m. and 6:00 p.m.? Considering the latitude of your study region and growth season, at that time it’s difficult for a foliar application to be effective, due to high solar radiation and high air temperature.

L107. What was the criterion selected for the fertilization of crop? The presented soil analysis shows acceptable levels (even high) of all the elements, especially N.

L112. Was the crop under irrigated or dried conditions? Please specify.

L162. I believe that the classification would be better proposed in the M&M section.

L221. The discussion can be improved, especially if the dynamics of the elements in the soil-plant system are discussed in greater depth.

In addition to the above, he suggested carrying out a correlation analysis between all the evaluated variables. It would be very interesting to verify if there is a relationship between SPAD values, photosynthesis, N, and yield. With this, the response of the plants to the applied treatments would be explained in a better way.

L285. The conclusion should be expanded and improved.

Reviewer #2: This manuscript describes the interaction between maize cultivars and Zn fertilization in Northern China. This is an important issue for agronomists around the world. This paper is suitable for publication in ´Plos One´. In general, the manuscript is worthy and well written. However, there are important issues that authors should correct, explain or expand before considering it for publication.

Title: In general is OK. The term ‘exogenous’ does not make sense.

Abstract: In the abstract authors affirmed that only 27.3% of maize cultivars responded positively to Zn fertilization, but in discussion section they stated that foliar ZnSO4 spray increased the grain yields of 18 maize cultivars, which accounted for 81.8% of the total cultivars. This sounds contradictory.

Keywords: OK

Introduction: In general is straightforward. However, authors should make a better effort to show the limitations/contradictions/deficiencies of previous research. As its present state, the motivation of the manuscript has a local/regional importance (not satisfactory results of northern China farmers for Zn application in maize). In this sense, the lack of hypotheses or research questions does not help. The specific objective 2 is not clear.

Material and methods: Some important aspects are incomplete described

Site Description: Level of mineral nitrogen (nitrates) in soil, type of extracted P (Olsen, Bray,M3?) and Zn (DPTA, M3?).

Experimental design: Accurate doses of Zn apply?.

Determinations: Please add some references.

Results: How did authors establish three different yield increase rate classes? How were the boundaries of the classes set?

Conclusion: I think it should response more specifically to the objectives of the research

References: I think the number is OK, but authors should increase the literature review, especially of those meta-analyses or reviews published in the last ten years (e.g. https://doi.org/10.3389%2Ffpls.2021.739282, https://doi.org/10.3389/fpls.2021.736658, https://doi.org/10.1016/j.heliyon.2023.e16040

Some specific comments

L54. Please replace ‘a plant´for ´plants’

L86. Please add ‘°C’ after ‘31.2’

L96. Please replace ‘species’ for ‘cultivars’.

L98. I think it is irrelevant the trademark of the maize cultivars.

L100. The zinc concentration is expressed in (w/v) but the total solution is expressed in w.

L101. 12?

L209. Please replace ‘between’ for ‘among’

L317. Please check the reference style.

Reviewer #3: Dear authors

Thank you for the opportunity to review this manuscript. The manuscript " Variations in grain yield and nutrient status of different maize cultivars by exogenous application of zinc sulphate" deals with an interesting topic and gives an insight into the potential of Zn application and yield improvement of maize. The strength of the manuscript lies in the fact that it discusses some widely used solutions in plant nutrition in maize production. The introduction provided a good insight into the topic analyzed and the research gaps. Considering the area and importance of maize, the study is of interest to a wider scientific audience. The selected topic is within the scope of the journal, literature is up to date. The evaluated paper is of an appropriate length and the language fully meets the publisher's requirements. The original idea is interesting, but it appears that the manuscript has not been successful in developing and verifying the aim of the study. The experimental setting can be better explained, and relationship between hybrid and treatment should be better establish to add merit to the current knowledge on maize nutrition. More detailed analyses are as follows and some specific comments have been made within the manuscript

Point 1. In the materials and methods more information of meteorological condition would be needed to introduce the readers with the suitability of the condition for maize growing. In addition to that what is altitude of the experimental site. The authors presented the soil properties for the 0-20cm soil depth. This could be sufficient for crops with shallow root system but to get more information on soil and specially Zn in soil deeper depth are also required.

Point 2 Experimental design needs further clarification i.e. e what was the main factor in this experiment is not clear how many replicates. Are there any effects of the years. The scheme with the experimental layout is also more than welcomed

Point 3. The question is why testing the Zn deficiency resistant hybrid to Zn doses and compare them with sensitive type.

Point 4. Does soil analyses clearly indicate low Zn availability or this is just an assumptions in this study. Please refer to line LINE 223.

Point 5. The literature needs technical arrangement and alignment

Point 6. Sequence of the tables should be organized in such way that first you showed the hybrid (Figure 1) and than

LINE 41-42: The main reason for the Zn deficiency is intensive farming and in this study you mention that extensive farming was used please explain? t could be also related with soil type please consider that also Alloway, B.J. Soil factors associated with zinc deficiency in crops and humans. Environ Geochem Health 31, 537–548 (2009). https://doi.org/10.1007/s10653-009-9255-4

LINE 57: ″farmers with enough time typically″ this is vague statement. Farmers are not hobbyist to do the farming in the free time please correct

LINE 95: What was the preceding crop and generally what is common type of cropping systems used in this experiment, tillage, seedbed preparation…What is mechanical sowing, which seeding machine was used

LINE 100-101: Please indicate how many liter of water was used per ha

LINE 108: ″10-leaf stage (45 d after sowing).″ from the agronomic point of view this is acceptable but for scientific study please use some common protocol to describe the growth stage such as BBCH scale or other.

LINE 135 -136: How those 5 plants were selected within the plots

LINE 162: ″The 22 maize cultivars were divided into three types″ is this separation done before or after the experimental set up

LINE 171 ″3.2. Yields and yield components of different maize types″ please correct to maize cultivars or hybrids

LINE 179: 100-grain weight change to 1000-grain weight – commonly this is reported in the studies of maze

LINE 226-227: The author reported yield increase in 81% of cultivars however this should be link to some average yield values or agrecological mean yield values in this study and supported with the statistical significance. In this way, there is no strong evidence that this is the result of the Zn application.

LINE 253-254: This can be also attributed to the root development and absorption capacity

6. PLOS authors have the option to publish the peer review history of their article (what does this mean?). If published, this will include your full peer review and any attached files.

Reviewer #1: No

Reviewer #2: No

Reviewer #3: No

---

## [Author Response · Author response to Decision Letter 0]

12 Sep 2023

Dear Editors and Reviewers,

Thank you very much for your advice and comments are relates to our manuscript. Your comments are very useful indeed toward the improvement of this draft. We have made the corresponding revisions according to the editors and reviewer’s comments. The English of the manuscript has also been further improved. The modifications made in the manuscript are marked in revised file. Our point-by-point responses to the comments of reviewers are included below as inserts marked in blue.

Thank you very much for considering our manuscript for publication in PLos One. I am indeed grateful for your time, and look forward to hearing from you soon!

Best wishes,

Dr. Juan Xin

Zhengzhou Normal University, Zhengzhou, 450044, China.

E-mail: xinjuan0707@163.com

Journal Requirements:

Response (R): Yes, we have revised our manuscript’s style as Journal requirements, thank you for your kind reminder.

R: We have supplemented this additional information ‘This field experiment was approved with Xvchang Agricultural Technology Extension Station’.

3. Thank you for stating the following financial disclosure: “The author(s) received no specific funding for this work.” At this time, please address the following queries:

R: We have supplemented our funding information in Funding section.

R: The funder is my work group, Zhengzhou Normal University, not people.

R: None.

R: We received funding from my work group, Zhengzhou Normal University.

R: Yes, we do.

4. Thank you for stating the following in the Acknowledgments Section of your manuscript: “The authors are thankful to the project, “Effects of stress on wheat seed germination and seedling growth metabolism with different nitrogen content” funded by the special fund for doctoral research startup, Zhengzhou Normal University for the study”. We note that you have provided additional information within the Acknowledgements Section that is not currently declared in your Funding Statement. Please note that funding information should not appear in the Acknowledgments section or other areas of your manuscript. We will only publish funding information present in the Funding Statement section of the online submission form. Please remove any funding-related text from the manuscript and let us know how you would like to update your Funding Statement. Currently, your Funding Statement reads as follows: “The author(s) received no specific funding for this work.” Please include your amended statements within your cover letter; we will change the online submission form on your behalf.

R: Get it, we have rewrote the Acknowledgements and Funding Section.

5. In your Data Availability statement, you have not specified where the minimal data set underlying the results described in your manuscript can be found. PLOS defines a study's minimal data set as the underlying data used to reach the conclusions drawn in the manuscript and any additional data required to replicate the reported study findings in their entirety. All PLOS journals require that the minimal data set be made fully available. For more information about our data policy, please see http://journals.plos.org/plosone/s/data-availability. Upon re-submitting your revised manuscript, please upload your study’s minimal underlying data set as either Supporting Information files or to a stable, public repository and include the relevant URLs, DOIs, or accession numbers within your revised cover letter. For a list of acceptable repositories, please see http://journals.plos.org/plosone/s/data-availability#loc-recommended-repositories. Any potentially identifying patient information must be fully anonymized. Important: If there are ethical or legal restrictions to sharing your data publicly, please explain these restrictions in detail. Please see our guidelines for more information on what we consider unacceptable restrictions to publicly sharing data: http://journals.plos.org/plosone/s/data-availability#loc-unacceptable-data-access-restrictions. Note that it is not acceptable for the authors to be the sole named individuals responsible for ensuring data access. We will update your Data Availability statement to reflect the information you provide in your cover letter.

R: We have upload our study’s minimal underlying data set upon re-submitting our revised paper.

R: We have upload our data set accompanied by our revised paper.

7. PLOS requires an ORCID iD for the corresponding author in Editorial Manager on papers submitted after December 6th, 2016. Please ensure that you have an ORCID iD and that it is validated in Editorial Manager. To do this, go to ‘Update my Information’ (in the upper left-hand corner of the main menu), and click on the Fetch/Validate link next to the ORCID field. This will take you to the ORCID site and allow you to create a new iD or authenticate a pre-existing iD in Editorial Manager. Please see the following video for instructions on linking an ORCID iD to your Editorial Manager account: https://www.youtube.com/watch?v=_xcclfuvtxQ

R: We have registered our ORCID iD (0009-0003-6693-9440).

8. Please upload a copy of Supplementary Table S1 which you refer to in your text on page 5.

R: We have uploaded the Table S1.

Editor Comments:

The manuscript requires additional work before being considered for possible acceptance. In particular, the reviewers emphasize the need to improve the Materials and Methods and Discussion sections. They also point out specific issues that must be resolved in discussing the results.

R: Thank you very much for providing us with an opportunity to revise this paper. Two reviewers put forward many good proposals on revision.

Reviewer Comments:

Reviewer 1: 

The manuscript shows interesting results that may be useful for the scientific community and for corn farmers. However, I believe that some aspects of it should be improved. Here are some recommendations:

R: Thank you for your attention and suggestions on our manuscript, we will carefully revise to reach the publication level.

1. L28. The keywords must be different from those of the title.

R: Yes, you are right, we have revised these keywords.

2. L38. Does what is mentioned occur in all types of soil?

R: This sentence is not very precise, not all soils are deficient in microelements. Generally, poor soil and soil that has not been applied organic fertilizer are easy to lack microelements.

3. L44. Mainly in alkaline pH soils.

R: Yes, higher pH in Northern China greatly reduces the availability of zinc in soil, resulting in zinc deficiency in plants. We have revised this sentence as your comments, thank you.

4. L87. Why is it thought that the depth of soil sample? Consider that the roots of corn explore much more than 20 cm.

R: Yes, you are right, the roots length of maize is much more than 20cm in field. The 20cm soil can only supply nutrients in the early stage of maize. But, this has basically become a default, except tree, the majority of soils which used for physical and chemical properties test come from 0-20cm / 0-30 cm depth soil.

Related reference:

(1) Aziiba EA, Hu FL, Fan ZL, Chai Q. Optimized nitrogen rate, plant density, and regulated irrigation improved grain, biomass yields, and water use efficiency of maize at the oasis irrigation region of China. Agriculture, 2022, 12, 234.

(2) Liu YX, Pan YQ, Yang L, Ahmad S, Zhou XB. Stover return and nitrogen application affect soil organic carbonand nitrogen in a double-season maize field. Plant Biology, 2022, 24: 387-395.

(3) Li J, Lv SQ, Yang ZY, Wang XF, Li HT, Bai YH, Zhou CH, Wang LQ, Abdo AI. Improving spring maize yield while mitigating nitrogen losses under film mulching system by right fertilization and planting placement. Field Crops Research, 2023, 290:108743.

5. L103. Are foliar applications done between 5:00 p.m. and 6:00 p.m.? Considering the latitude of your study region and growth season, at that time it’s difficult for a foliar application to be effective, due to high solar radiation and high air temperature.

R: The maize growing season is very hot, especially between 12:00 and 15:00, field temperatures can exceed 38℃. At 5:00 p.m.- 6:00 p.m (17:00-18:00), field temperatures can drop by 3-4℃. The air temperature is relatively low and the solar radiation gradually decreases into the evening. At this time, the evaporation of foliar fertilizer was smaller, and the adhesion agent (Tween-20 [(polyethylene glycol sorbitan monolaurate); Aladdin Industrial Corporation, Shanghai, China]) was added to strengthen the viscosity of foliar fertilizer, which effectively reducing evaporation and ensuring the effect of foliar fertilizer.

6. L107. What was the criterion selected for the fertilization of crop? The presented soil analysis shows acceptable levels (even high) of all the elements, especially N.

R: Thank you for your careful review. The soil K is slightly higher, OM and N is medium, P is low, Zn is very low. Be honestly, we have consulted experts from Henan Agricultural University, who have done fertilizer rate tests locally and told us that the N: P2O5: K2O=180: 90: 90 is reasonable. We cited their literature in our revised paper.

Index Organic matter Total N Available P Available K available Zn pH

Value 20.03 g kg–1 1.45 g kg–1 18.67 mg kg–1 157.32 mg kg–1 0.40 mg kg–1 7.8

Degree of abundance Medium Medium Low Slightly higher Very low Alkaline

7. L112. Was the crop under irrigated or dried conditions? Please specify.

R: Thank you for your suggestion. The maize was grown under a irrigated field, we supplemented the meteorological data and irrigation condition. Due to sufficient rainfall, we did not carry out additional irrigation during the maize growing season.

8. L162. I believe that the classification would be better proposed in the M&M section.

R: Yes, your suggestion sounds good, thank you. We have put this classification in the Materials and Methods section.

9. L221. The discussion can be improved, especially if the dynamics of the elements in the soil-plant system are discussed in greater depth. In addition to the above, he suggested carrying out a correlation analysis between all the evaluated variables. It would be very interesting to verify if there is a relationship between SPAD values, photosynthesis, N, and yield. With this, the response of the plants to the applied treatments would be explained in a better way.

R: We added the dynamics of Zn in the soil-plant system in the scetion of Discussion. We added the correlation graph in the revision, as you’d expect, this figure shows lots of useful information to help us better understand the mechanism by which zinc enhances grain yield.

Figure 8 Correlation between photosynthetic characteristics, nutrient uptake and yield.

10. L285. The conclusion should be expanded and improved.

R: We have revised our conclusion, thank you.

Reviewer 2: 

This manuscript describes the interaction between maize cultivars and Zn fertilization in Northern China. This is an important issue for agronomists around the world. This paper is suitable for publication in ´Plos One´. In general, the manuscript is worthy and well written. However, there are important issues that authors should correct, explain or expand before considering it for publication.

R: Thank you for your patient guidance, and we appreciate your approval of our manuscript’s opinion.

Title: In general is OK. The term ‘exogenous’ does not make sense.

R: This word had been removed.

Abstract: In the abstract authors affirmed that only 27.3% of maize cultivars responded positively to Zn fertilization, but in discussion section they stated that foliar ZnSO4 spray increased the grain yields of 18 maize cultivars, which accounted for 81.8% of the total cultivars. This sounds contradictory.

R: The 81.8% including Type S (27.3%) and Type N (54.5%), but you're right, the yield increase in Type N maize was not obvious. We have changed ‘81.8%’ to ’27.3’.

Keywords: OK

R: Thank you.

Introduction: In general is straightforward. However, authors should make a better effort to show the limitations/contradictions/deficiencies of previous research. As its present state, the motivation of the manuscript has a local/regional importance (not satisfactory results of northern China farmers for Zn application in maize). In this sense, the lack of hypotheses or research questions does not help. The specific objective 2 is not clear.

R: Your suggestions are very important. We supplemented some information of maize Zn sensitivity in the section of Introduction, and revised the objective 2, thank you.

Material and methods: Some important aspects are incomplete described Site Description: Level of mineral nitrogen (nitrates) in soil, type of extracted P (Olsen, Bray, M3?) and Zn (DPTA, M3?).

Experimental design: Accurate doses of Zn apply?. Determinations: Please add some references.

R: Thank you for your suggestion, we supplemented these important information (Soil Nmin, Olsen-P, NH4OAc-K, DTPA-Zn) and related rederences in our revised paper.

Results: How did authors establish three different yield increase rate classes? How were the boundaries of the classes set?

R: The between-groups linkage method in hierachical chuster analysis (SPSS19.0 software, Chicago, IL, USA) was used for maize variety classification, with an interval of Squared Euclidean distance. We supplemented this information in the section of Materials and Methods (maize type classification).

Conclusion: I think it should response more specifically to the objectives of the research.

R: We have revised the conclusions to align them with the objective, thank you.

References: I think the number is OK, but authors should increase the literature review, especially of those meta-analyses or reviews published in the last ten years (e.g. https://doi.org/10.3389%2Ffpls.2021.739282, https://doi.org/10.3389/fpls.2021.736658, https://doi.org/10.1016/j.heliyon.2023.e16040

R: Thank you for your suggestion, we have read more latest literatures and added them in References.

Some specific comments:

1. L54. Please replace ‘a plant´for ´plants’

R: Ok, we have replaced it.

2.L86. Please add ‘°C’ after ‘31.2’

R: This had been added, thank you for your kindly reminder.

L96. Please replace ‘species’ for ‘cultivars’.

R: Yes, ‘cultivars’ is correct. 

L98. I think it is irrelevant the trademark of the maize cultivars.

R: Yes, it can be ignored, we have removed the company that bought the maize seeds.

L100. The zinc concentration is expressed in (w/v) but the total solution is expressed in w.

R: Yes, the unit is not clear here, we have revised it. Changed as “one was sprayed with 1350 g ZnSO4•7H2O at 450 L solution per hectare (+Zn)”. The concentration of zinc fertilizer is 3 g/L.

L101. 12?

R: Sorry, it is [12], a reference. This had been revised.

L209. Please replace ‘between’ for ‘among’

R: Done as your suggestion, thank you.

L317. Please check the reference style.

R: Thank you for your kindly suggestion, we have checked them all and revised some errors.

Reviewer 3:

Thank you for the opportunity to review this manuscript. The manuscript " Variations in grain yield and nutrient status of different maize cultivars by exogenous application of zinc sulphate" deals with an interesting topic and gives an insight into the potential of Zn application and yield improvement of maize. The strength of the manuscript lies in the fact that it discusses some widely used solutions in plant nutrition in maize production. The introduction provided a good insight into the topic analyzed and the research gaps. Considering the area and importance of maize, the study is of interest to a wider scientific audience. The selected topic is within the scope of the journal, literature is up to date. The evaluated paper is of an appropriate length and the language fully meets the publisher's requirements. The original idea is interesting, but it appears that the manuscript has not been successful in developing and verifying the aim of the study. The experimental setting can be better explained, and relationship between hybrid and treatment should be better establish to add merit to the current knowledge on maize nutrition. More detailed analyses are as follows and some specific comments have been made within the manuscript.

R: Thank you very much for providing us with an opportunity to revise our paper. I could feel your professionalism in the process of modifying this manuscript.

Point 1. In the materials and methods more information of meteorological condition would be needed to introduce the readers with the suitability of the condition for maize growing. In addition to that what is altitude of the experimental site. The authors presented the soil properties for the 0-20cm soil depth. This could be sufficient for crops with shallow root system but to get more information on soil and specially Zn in soil deeper depth are also required.

R: Yes, we had added the altitude, temperature and rainfall data in Materials and Methods.

Yes, this data (Zn content of deeper depth soil) is very important because the root system of maize can grow up to 80-100 cm, and its main root is concentrated in 40-60 cm. However, unfortunately, we did not take the deeper soil at that time.

Point 2 Experimental design needs further clarification i.e. e what was the main factor in this experiment is not clear how many replicates. Are there any effects of the years. The scheme with the experimental layout is also more than welcomed.

R: In order to facilitate readers to understand the experimental design, we drew a figure of the plot distribution as your suggestion.

Point 3. The question is why testing the Zn deficiency resistant hybrid to Zn doses and compare them with sensitive type.

R: 22 varieties were classified into different zinc-sensitive types. This study compared their responses to zinc fertilizer application in terms of agronomic traits, aging, photosynthesis, and yield. The results indicate that zinc fertilizer should not be sprayed solely due to soil zinc deficiency, and varieties should also be considered. Excessive zinc fertilizer would inhibit the absorption of P and K, accelerate leaf senescence and result in yield reduction.

Point 4. Does soil analyses clearly indicate low Zn availability or this is just an assumptions in this study. Please refer to line LINE 223.

R: Yes, this is just an assumption for various plants under Zn-deficient soils.

Point 5. The literature needs technical arrangement and alignment.

R: Yes, the previous format of literature does not meet the requirements of the journal, and we have re-arranged them.

Point 6. Sequence of the tables should be organized in such way that first you showed the hybrid (Figure 1) and than

R: Dear reviewer and editor, this sentence does not seem to finish, there is some bug when uploading? I am very sorry, I don't understand the meaning of this sentence.

1. LINE 41-42: The main reason for the Zn deficiency is intensive farming and in this study you mention that extensive farming was used please explain? t could be also related with soil type please consider that also

Alloway, B.J. Soil factors associated with zinc deficiency in crops and humans. Environ Geochem Health 31, 537–548 (2009). https://doi.org/10.1007/s10653-009-9255-4

R: Thank you for your sharing, we have read this classic paper. In Northern China, the main soil factors affecting the availability of Zn to plants are higher pH. In addition to environmental factors, the human factor is intensive farming without paying attention to supplement microelement (few farmers are willing to apply organic fertilizer/microelement fertilizer on cereal crop, resulting in soil zinc deficiency). We have revised this sentence in revised paper, thank you.

2. LINE 57: ″farmers with enough time typically″ this is vague statement. Farmers are not hobbyist to do the farming in the free time please correct.

R: Yes, you are right, we have revised this sentence.

3. LINE 95: What was the preceding crop and generally what is common type of cropping systems used in this experiment, tillage, seedbed preparation…What is mechanical sowing, which seeding machine was used.

R: The preceding crop is wheat, and the field sites took the wheat–maize rotation system during experimental period. No tillage, no seedbed, we applied the maize seed and fertilizer at the same time through spoon wheel type maize simple grain sowers (2BYFSF-4, Hebei Nonghaha Machinery Group Co.,Ltd, Shijiazhuang, China). We supplemented these information in M&M section.

4. LINE 100-101: Please indicate how many liter of water was used per ha.

R: Yes, this information is lacking and we had supplemented it in M&M section. We used 30 L water per 666.7 m2 = 450 L/ha.

5. LINE 108: ″10-leaf stage (45 d after sowing).″ from the agronomic point of view this is acceptable but for scientific study please use some common protocol to describe the growth stage such as BBCH scale or other.

R: We have changed them all to BBCH model, thank you.

6. LINE 135 -136: How those 5 plants were selected within the plots.

R: We have supplemented this information. 5 consecutive maize plants were selected from each plot. Commonly, the growth of maize in the field is relatively uniform

7. LINE 162: ″The 22 maize cultivars were divided into three types″ is this separation done before or after the experimental set up.

R: The 22 maize varieties were separated into three categories after the experiment set up. Before the experimental set up, we already know that the local soil is zinc deficiency. We initially believed that these maize varieties would have a significant increase in yield after spraying zinc fertilizer, but the experimental results were not what we expected.

8. LINE 171 ″3.2. Yields and yield components of different maize types″ please correct to maize cultivars or hybrids.

R: Yes, ‘maize types’ changed as ‘maize cultivars’, others had been changed accordingly.

9. LINE 179: 100-grain weight change to 1000-grain weight – commonly this is reported in the studies of maize.

R: Yes, we have revised ‘100-grain weight’ to ‘1000-grain weight’, and updated the data in Table 2.

10. LINE 226-227: The author reported yield increase in 81% of cultivars however this should be link to some average yield values or agrecological mean yield values in this study and supported with the statistical significance. In this way, there is no strong evidence that this is the result of the Zn application.

R: Yes, you are right. We compared the yield differences between +Zn and -Zn treatments for each variety and marked their significances (Fig. 3). The results showed that the yield of 45.5% varieties increased significantly after applying zinc fertilizer, but the yield of 13.6% varieties decreased significantly.

11. LINE 253-254: This can be also attributed to the root development and absorption capacity.

R: There have been lots of studies on the inhibition of zinc absorption by high phosphorus input in plants, and the mechanism of inhibiting zinc absorption and transport by high P is also clear. A few studies reported that higher Zinc inhibited the P uptake by roots, and the physiological mechanism is not clear.

---

## [Decision Letter · Decision Letter 1]

26 Oct 2023

PONE-D-23-07689R1Variations in grain yield and nutrient status of different maize cultivars by exogenous application of zinc sulphatePLOS ONE

Dear Dr. Xin,

Thank you for submitting your manuscript to PLOS ONE. After careful consideration, we feel that it has merit but does not fully meet PLOS ONE’s publication criteria as it currently stands. Therefore, we invite you to submit a revised version of the manuscript that addresses the points raised during the review process.

The manuscript was substantially improved. However, it still requires some adjustments to ensure the quality required.

The reviewers indicate the following points:

(1). insert the size of the experiment as well as the dimension of the elementary plot.

(2). LINE 163-164: This sentence doesn't have a clear meaning.

(3). LINE 228 "....yields and grain numbers of different maize. " What grain numbers refer to in this sentence here and after that?

Check the English language of the manuscript (please see corrections in the attached file).

Table 1 repeats the information described in the manuscript's text (please choose one option: Table or text).

Please check the references to homogenize and adjust to the requested style.

We look forward to receiving your revised manuscript.

Kind regards,

Adalberto Benavides-Mendoza, Ph.D.

Academic Editor

PLOS ONE

Journal Requirements:

Additional Editor Comments:

The manuscript was substantially improved. However, it still requires some adjustments to ensure the quality required.

The reviewers indicate the following points:

(1). insert the size of the experiment as well as the dimension of the elementary plot.

(2). LINE 163-164: This sentence doesn't have a clear meaning.

(3). LINE 228 "....yields and grain numbers of different maize. " What grain numbers refer to in this sentence here and after that?

Check the English language of the manuscript (please see corrections in the attached file).

Table 1 repeats the information described in the manuscript's text (please choose one option: Table or text).

Please check the references to homogenize and adjust to the requested style.

Reviewers' comments:

Reviewer's Responses to Questions

**Comments to the Author**

1. If the authors have adequately addressed your comments raised in a previous round of review and you feel that this manuscript is now acceptable for publication, you may indicate that here to bypass the “Comments to the Author” section, enter your conflict of interest statement in the “Confidential to Editor” section, and submit your "Accept" recommendation.

Reviewer #1: All comments have been addressed

Reviewer #2: All comments have been addressed

Reviewer #3: All comments have been addressed

2. Is the manuscript technically sound, and do the data support the conclusions?

Reviewer #1: Yes

Reviewer #2: Yes

Reviewer #3: Yes

3. Has the statistical analysis been performed appropriately and rigorously? 

Reviewer #1: Yes

Reviewer #2: Yes

Reviewer #3: Yes

4. Have the authors made all data underlying the findings in their manuscript fully available?

Reviewer #1: Yes

Reviewer #2: Yes

Reviewer #3: Yes

5. Is the manuscript presented in an intelligible fashion and written in standard English?

Reviewer #1: Yes

Reviewer #2: No

Reviewer #3: Yes

6. Review Comments to the Author

Reviewer #1: The authors have addressed all the recommendations provided. The article improved substantially. I suggest acceptance to be published in this journal.

Reviewer #2: The manuscript has been improved following reviewers comments. I suggest checking the English of the manuscript (please see corrections in the attached file). I think that Table 1 repeats information described in the text of the manuscript (please choose the one option). Please check the references.

Reviewer #3: Dear authors

Thank you for the detailed and clear answers to my questions, as well as the responses to the recommendations and suggestions of the other reviewers. In this round, the paper has been significantly improved and the overall research has gained in flow and results significance. I am grateful to the authors for their efforts to answer the many questions raised by the reviewers.

After re-reading and analyzing, I would have one recommendation, which is to clearly insert the size of the experiment as well as the dimension of the elementary plot. Also two small suggestions

LINE 163-164: This sentence doesn have clear meaning PLs correct

LINE 228 "....yields and grain numbers of different maize. " What grain numbers refers to in this sentence here and thereafter

7. PLOS authors have the option to publish the peer review history of their article (what does this mean?). If published, this will include your full peer review and any attached files.

Reviewer #1: No

Reviewer #2: No

Reviewer #3: No

---

## [Editor Report · Decision Letter 2]

22 Nov 2023

Variations in grain yield and nutrient status of different maize cultivars by application of zinc sulfate

PONE-D-23-07689R2

Dear Dr. Xin,

We’re pleased to inform you that your manuscript has been judged scientifically suitable for publication and will be formally accepted for publication once it meets all outstanding technical requirements.

Kind regards,

Adalberto Benavides-Mendoza, Ph.D.

Academic Editor

PLOS ONE

Additional Editor Comments (optional):

The authors resolved all the concerns and suggestions of the reviewers. Therefore, the manuscript can be accepted for publication.
---

## [Editor Report · Acceptance letter]

28 Nov 2023

PONE-D-23-07689R2 

Variations in grain yield and nutrient status of different maize cultivars by application of zinc sulfate 

Dear Dr. Xin:

I'm pleased to inform you that your manuscript has been deemed suitable for publication in PLOS ONE. Congratulations! Your manuscript is now with our production department. 

Kind regards, 

on behalf of

Dr. Adalberto Benavides-Mendoza 

Academic Editor

PLOS ONE